# Inflammatory Glycoprotein YKL-40 Is Elevated after Coronary Artery Bypass Surgery and Correlates with Leukocyte Chemotaxis and Myocardial Injury, a Pilot Study

**DOI:** 10.3390/cells11213378

**Published:** 2022-10-26

**Authors:** Antti Laurikka, Katriina Vuolteenaho, Vesa Toikkanen, Timo Rinne, Tiina Leppänen, Mari Hämäläinen, Matti Tarkka, Jari Laurikka, Eeva Moilanen

**Affiliations:** 1The Immunopharmacology Research Group, Faculty of Medicine and Health Technology, Tampere University Hospital, Tampere University, 33014 Tampere, Finland; 2Tampere University Hospital Heart Center Co., P.O. Box 2000, 33521 Tampere, Finland; 3Department of Anaesthesia, Tampere University Hospital, P.O. Box 2000, 33521 Tampere, Finland; 4Finnish Cardiovascular Research Center Tampere, Tampere University, 33014 Tampere, Finland

**Keywords:** YKL-40, cardiac surgery, ischemia-reperfusion injury, myocardial injury, chemotactic factors

## Abstract

The aim of the present study was to investigate the levels of YKL-40 during and after coronary artery bypass grafting surgery (CABG) and to establish possible connections between YKL-40 and markers of oxidative stress, inflammation, and myocardial injury. Patients undergoing elective CABG utilizing cardiopulmonary bypass (CPB) were recruited into the study. Blood samples were collected at the onset of anesthesia, during surgery and post-operatively. Levels of YKL-40, 8-isoprostane, interleukin-8 (IL-8), monocyte chemotactic protein-1 (MCP-1) and troponin T (TnT) were measured by immunoassay. YKL-40 levels increased significantly 24 h after CPB. Positive correlation was seen between post-operative TnT and YKL-40 levels (r = 0.457, *p* = 0.016) and, interestingly, baseline YKL-40 predicted post-operative TnT increase (r = 0.374, *p* = 0.050). There was also a clear association between YKL-40 and the chemotactic factors MCP-1 (r = 0.440, *p* = 0.028) and IL-8 (r = 0.484, *p* = 0.011) linking YKL-40 to cardiac inflammation and fibrosis following CABG. The present results show, for the first time, that YKL-40 is associated with myocardial injury and leukocyte-activating factors following coronary artery bypass surgery. YKL-40 may be a factor and/or biomarker of myocardial inflammation and injury and subsequent fibrosis following heart surgery.

## 1. Introduction

Coronary artery disease (CAD) often requires invasive therapy, i.e., percutaneous coronary invasion (PCI) or coronary artery bypass grafting surgery (CABG). CABG utilizing cardiopulmonary bypass (CPB) is an effective treatment but it is complicated with systemic inflammatory response syndrome (SIRS), ischemia-reperfusion syndrome and morbidity [1]. For instance, preoperative LDL-levels and postoperative rehabilitation have been shown to improve prognosis and recovery from heart surgery [2,3]. Biomarker that could recognize patients with myocardial injury and poor prognosis could be utilized to focus therapeutic interventions to high-risk patients.

YKL-40 is a chitin-binding glycoprotein, known also as chitinase-3-like protein 1 (CHI3L1) describing the lack of chitinase activity that characterizes true chitinases. YKL-40 is expressed in many cell types, including neutrophils, macrophages, and vascular smooth muscle cells [4,5] as well as chondrocytes and cancer cells [5,6]. YKL-40 serves as growth, adhesion and migration factor having a role in proliferation and fibrosis of connective tissue [5,7,8]. Accordingly, associations of YKL-40 with inflammation and tissue remodeling in cancer, liver and lung fibrosis, arthritis and strenuous joint loading have been reported [9,10,11,12,13,14]. The function of YKL-40 is still, however, largely unknown. In the framework of the present study, it is interesting that YKL-40 has been shown to increase in acute myocardial infarction [15,16] and in chronic heart failure [5,17]. In addition, increased levels of YKL-40 expression have been found in arteriosclerotic plaques suggesting that YKL-40 plays a role in plaque formation [5,18]. Poorer prognosis and increased rate of adverse cardiovascular events have been linked to high YKL-40 serum levels in patients with CAD or congestive heart disease (CHD) [16,19,20]. YKL-40 levels also increase with CAD severity [21,22]. YKL-40 has also been identified as an independent risk factor for all-cause mortality in patients acutely admitted to the hospital [23]. Interestingly, the expression of YKL-40 was increased in rat hearts recovering from ischemia-like insult of the myocardium with tissue remodeling when compared to continuously volume-overloaded hearts [24].

In the present study, YKL-40 and markers of myocardial injury (troponin T [TnT]), leukocyte activation (monocyte chemotactic protein-1 [MCP-1] and interleukin 8 [IL-8]) and oxidative stress (8-isoprostane) were analyzed in blood samples collected from patients undergoing elective CABG surgery. We hypothesized that YKL-40 may participate in the inflammatory cascade following open-heart surgery utilizing CPB and might function as a biomarker of inflammation and myocardial injury.

## 2. Materials and Methods

The present study included thirty-two patients (8 females, BMI 28.5 ± 0.8 kg/m^2^, age 68 ± 2 years; mean ± SEM) admitted to the Tampere University Hospital Heart Center Co to have an elective on-pump CABG with CPB. Canadian Cardiovascular Society (CCS) angina scale was used to assess the severity of coronary artery disease (CAD) and at least grade II symptoms were set as inclusion criteria. The average CCS value in these patients was grade III. Pre-existent pulmonary disease, pulmonary hypertension (PAP > 40 mmHg), smoking, impaired left ventricular function (LVEF < 50%), myocardial infarction, cardiac failure period or acute coronary syndrome during the previous 3 months, previous malignancy, infection and/or treatment with COX-2 selective nonsteroidal anti-inflammatory drugs or glucocorticoids during the previous month were used as exclusion criteria. Three patients had very high TnT levels (TnT > 1000 ng/L, and one of them had a documented myocardial infarction during the post-operative period) and one patient an aberrantly high MCP-1 levels post-operatively. These four patients were excluded from the final statistical analysis to minimize the confounding effects of post-operative complications. Anesthesia, perfusion, surgical approach, and intensive care unit protocol were performed according to standard procedure as previously described [25]. Patient characteristics are summarized in Table 1.

Blood samples were drawn from the radial artery. At the onset of anesthesia, the first sample was taken at the baseline (BL) and the second sample was collected at the onset of CPB (CPB). The third sample was acquired one minute after the removal of aortic cross-clamp (ACC + 1 min) and the fourth sample was drawn 15 min after the removal of aortic cross-clamp upon reinflating lungs (ACC + 15 min). The last two samples were taken 4 and 24 h after the onset of CPB (CPB + 4 h and CPB + 24 h).

Obtained aliquots of plasma were kept at −80 °C until analysis. Concentrations of YKL-40, MCP-1, IL-8 and 8-isoprostane in the plasma samples were determined by immunoassays using following reagents: YKL-40 and MCP-1 (DuoSet ELISA, R&D Systems Europe Ltd., Abingdon, UK), IL-8 (BD Biosciences, Erembodegem, Belgium), and 8-isoprostane (Cayman, Ann Arbor, MI, USA). HUSLAB (Helsinki University Hospital, Helsinki, Finland) measured troponin T (TnT) concentrations by using immunochemiluminometric method validated for clinical use.

Data are shown as mean ± standard error of mean (SEM) or as indicated otherwise. According to data distribution, Spearman’s and Pearson’s correlations were used. A logarithmic transformation was used to create normal distribution when needed. Correlation was detected when R-values were greater than 0.3 and −0.3 [26]. Area under the curve (AUC) was calculated using trapezoid method with Graphpad Prism 8.0.0 (GraphPad Software, Inc., La Jolla, CA, USA). Statistical significance of variations in cytokine levels between time points was tested by repeated measures ANOVA and Bonferroni post hoc test using IBM SPSS statistics 22.0 (IBM, Armonk, NY, USA).

## 3. Results

### 3.1. YKL-40 Levels Increased Significantly after CABG Surgery

Mean baseline level of YKL-40 was 36 ± 5 ng/mL. During surgery, YKL-40 levels remained relatively constant. Post-operatively, four hours after onset of CPB, YKL-40 concentrations were still close to the baseline levels. Unexpectedly, YKL-40 levels elevated significantly 24 h after the CPB onset to a mean level of 1720 ± 205 ng/mL (Figure 1a).

### 3.2. YKL-40 Correlated with TnT, A Marker of Myocardial Injury

A clear increase in TnT levels was seen during surgery (Figure 1e). Interestingly, baseline YKL-40 levels correlated positively with TnT levels measured 24 h after the onset of surgery (r = 0.374, *p* = 0.050, Figure 2a, Table 2). YKL-40 measured at the same 24 h timepoint with TnT levels correlated even more strongly (r = 0.457, *p* = 0.016, Figure 2b, Table 2).

### 3.3. YKL-40 Correlated with Chemotactic Cytokines MCP-1 and IL-8

MCP-1 and IL-8 levels started to increase early during surgery and peaked at 4 h after the onset of CPB (Figure 1b,c), i.e., before any increase in YKL-40 was seen. MCP-1 levels had normalized 24 h after the onset of CPB, whereas IL-8 levels were still significantly elevated. Interestingly, AUC of MCP-1 was significantly associated with the peak levels of YKL-40 (r = 0.440, *p* = 0.028, Figure 3a, Table 2). Furthermore, IL-8 concentrations showed correlation with YKL-40 at baseline (r = 0.500, *p* = 0.007, Table 2) and at 24 h (r = 0.484, *p* = 0.011, Figure 3b).

8-isoprostane levels also increased early during surgery as a marker of oxidative stress but values were normalized by 4 h after CPB (Figure 1d). No significant correlations were found between YKL-40 and 8-isoprostane levels. Based on these results, YKL-40 seems to be associated with TnT release and levels of leukocyte-activating factors but not with oxidative stress.

## 4. Discussion

Our main finding was that YKL-40 levels greatly elevated after CABG surgery, and patients with increased release of the myocardial injury marker TnT also had higher post-operative YKL-40 concentrations. In addition, YKL-40 correlated with the chemotactic cytokines MCP-1 and IL-8 but not with 8-isoprostane, a biomarker of oxidative stress. This is the first study to measure YKL-40 levels during and after CABG surgery, and our findings are in line with the knowledge on biological functions of YKL-40 and with studies on cardiac patients.

Our results are supported by Harutyunyan et al. showing 25% increase in YKL-40 levels the day after percutaneous coronary intervention (PCI) in patients with CAD [27]. In our study, CABG procedure increased YKL-40 concentrations nearly 50-fold and the YKL-40 levels correlated positively with myocardial injury. Use of cardioplegia has made CABG surgery safer, but tissue injury can still occur after myocardial perfusion has been re-established [1,28]. The injury can range from slightly impaired myocardial contractility called myocardial stunning to contraction band necrosis and irreversible damage [29]. Ischemic injury and subsequent reperfusion cause influx of leukocytes into myocardium. This inflammatory response causes additional tissue injury through release of reactive oxygen species (ROS), release of matrix metalloproteinases and inflammatory mediators that strengthen the inflammatory response [29]. Neutrophils peak at 24 h after ischemic injury, and afterwards monocytes become the most prominent leukocyte type found in injured area [30].

Our data show that circulating MCP-1 and IL-8 correlate with plasma YKL-40 levels. MCP-1 is a potent chemotactic cytokine produced by inflammatory cells [31]. MCP-1 is expressed also in myocardium rapidly after myocardial infarction [32]. In the present study, MCP-1 levels rose early during surgery, peaked four hours after the onset of CPB and fell towards baseline levels within 24 h after the onset of CPB, supporting the results reported by Kortekaas et al. [33]. In the present study, the peak levels of MCP-1 were detected before YKL-40 levels were increased and, interestingly, the AUC of MCP-1 concentrations correlated positively with the maximum increase of YKL-40.

IL-8 (also known as chemokine CXCL8) is a potent chemotactic agent produced primarily by monocytes and IL-8 release results in recruitment of neutrophils into injured myocardium [34]. IL-8 levels increased significantly after reperfusion and stayed elevated 24 h after the onset of CPB. After myocardial ischemia, monocytes of splenic origin are recruited into blood stream [35]. Since IL-8 is primarily produced by monocytes, the substantial increase in IL-8 levels found in the present study may be secondary to mobilization of splenic monocytes. YKL-40 may also stimulate IL-8 production as shown by Tang et al. [36]. Association between YKL-40 and IL-8 is supported by the current findings on clear positive correlations between these two factors at baseline and especially post-operatively. It is also known that YKL-40 is produced by macrophages [5], so taking our findings into account, it is possible that a major portion of the significant rise in YKL-40 levels post-operatively is released from leukocytes activated and recruited by chemokines such as MCP-1 and IL-8.

In the present study, 8-isoprostane levels increased significantly at the onset of CPB and returned to baseline levels post-operatively. CPB has been shown to cause oxidative stress due to the inflammatory response it elicits and the high arterial partial oxygen pressures used during CPB [37]. As 8-isoprostane is a product of arachidonic acid oxidation following reactive oxygen species (ROS) release [38], we believe that the increase in 8-isoprostane levels after the onset of CPB is caused by the hyperoxic conditions during CPB and the ROS released secondary to the inflammatory response to CPB.

YKL-40 has been shown to stimulate connective tissue growth and thus may contribute to tissue fibrosis [5,7]. YKL-40 also inhibits cellular responses to interleukin-1 and tumor necrosis factor α resulting in reduced MMP-1, MMP-3 and MMP-13 release [39]. Thus, the marked increase in YKL-40 levels post-operatively may contribute to the development of myocardial fibrosis secondary to myocardial injury. If this is the case, it would be seminal to investigate whether high YKL-40 levels predict development of cardiac fibrosis and impaired left ventricular function post-operatively, as YKL-40 has been linked with poor prognosis among CAD and CHD patients [16,19,20]. The findings reported by Huuskonen et al. support this assumption. In that study, rat hearts recovering from acute volume overload-induced myocardial insult with tissue remodeling showed higher YKL-40 expression when compared to continuously volume overloaded-hearts [24].

The present study was a pilot study looking for changes in YKL-40 and cytokine levels during and after CABG surgery in order to generate hypothesis and research plan for further studies. As a pilot study, we limited the follow-up time to the intra-operative period and up to the first post-operative morning as cytokine levels normally taper off after the first post-operative day following CABG using CPB [40]. In addition, we assumed that possible complications and conditions appearing during the post-surgical period such as myocardial infarctions or infections might confound the data. However, YKL-40 levels were still on the increase at the last 24 h timepoint. In future studies, follow-up time should be extended at least up to the fourth post-operative day to follow YKL-40 levels during the recovery phase. Another limitation is the small patient cohort (*n* = 32) in which we decided to exclude four patients with aberrantly high TnT or MCP-1 levels from the final analysis to minimize the confounding effect of a post-operative complications and to exclude the risk of type I error. All patients included, the correlation between YKL-40 and TnT would have been even greater (r = 0.518, *p* = 0.003, *n* = 32).

In conclusion, we report here, for the first time, that YKL-40 levels significantly increase following open-heart surgery and correlate with markers of inflammation and myocardial injury. Thus, YKL-40 may be an effector and biomarker of myocardial inflammation and injury and subsequent remodeling following heart surgery. These findings warrant further research on the effects of YKL-40 during and after myocardial injury and on the impact of increased YKL-40 levels on cardiac function, morbidity and mortality following coronary artery bypass surgery.

## Figures and Tables

**Figure 1 cells-11-03378-f001:**
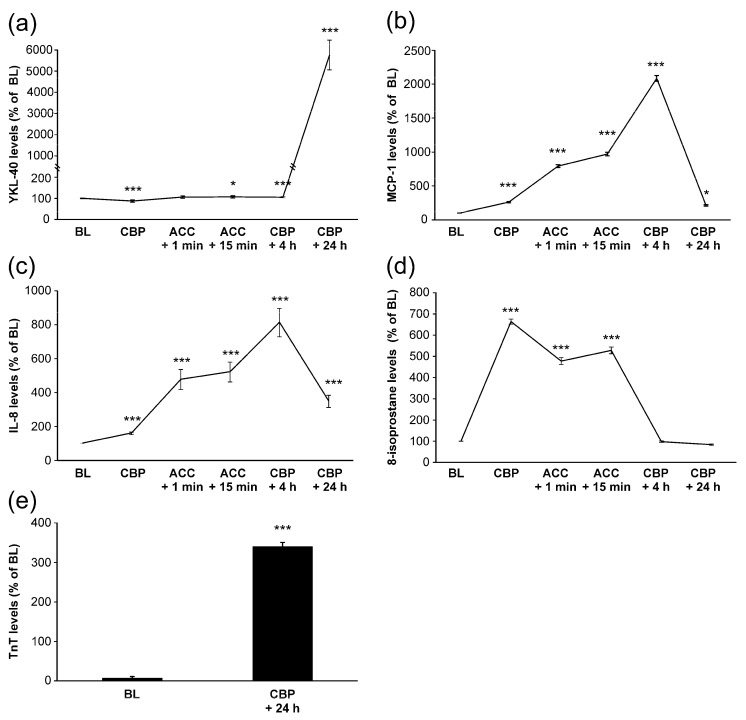
Levels of YKL-40 (**a**), monocyte chemotactic protein-1 (MCP-1, (**b**)), interleukin-8 (IL-8, (**c**)), 8-isoprostane (**d**) and troponin T (TnT, (**e**)) during and after coronary artery bypass grafting operation. Samples were drawn from the radial artery at the following time points: onset of anesthesia (baseline, BL), onset of cardiopulmonary bypass (CPB), 1 min after removal of aortic cross clamp (ACC + 1 min), 15 min after removal of ACC (ACC + 15 min), 4 h after onset of CPB (CPB + 4 h) and 24 h after onset of CPB (CPB + 24 h). Results are presented as mean + SEM, *n* = 28 patients. * = *p* < 0.05, *** = *p* < 0.001.

**Figure 2 cells-11-03378-f002:**
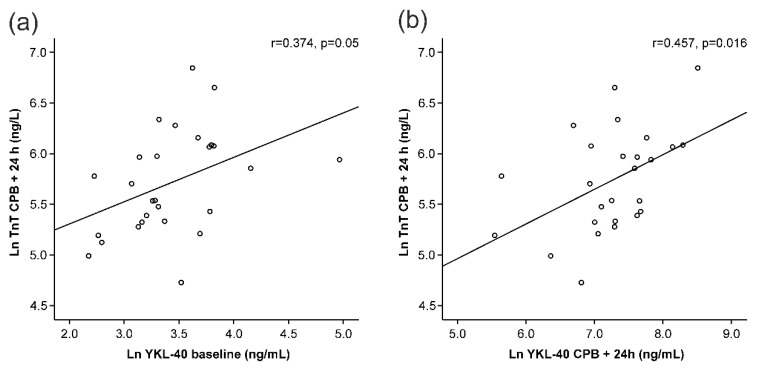
Correlations of troponin T (TnT) levels 24 h after the onset of cardiopulmonary bypass (CPB) to YKL-40 at baseline (**a**) and 24 h after the onset of CPB (**b**). All variables are presented in logarithmic form to obtain normal distributions. Pearson’s correlation was calculated (r). *p* < 0.05 was considered statistically significant.

**Figure 3 cells-11-03378-f003:**
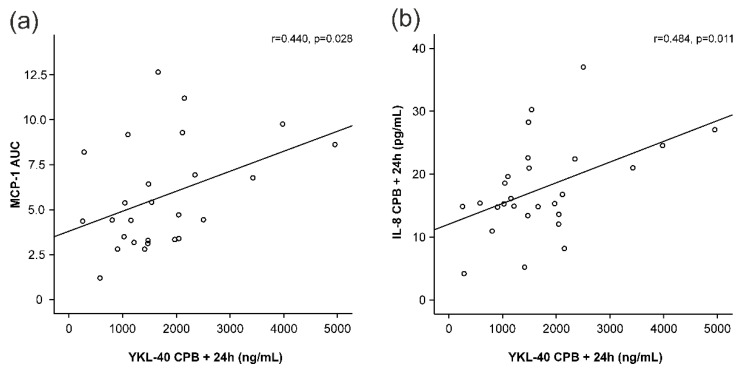
Correlations of peak YKL-40 levels (YKL-40 levels 24 h after the onset of cardiopulmonary bypass, CPB) to areas under curve (AUC) of monocyte chemotactic protein-1 (MCP-1, (**a**)) and interleukin-8 (IL-8, (**b**)) levels 24 h after the onset of cardiopulmonary bypass (CPB). Spearman’s correlation was calculated (r). *p* < 0.05 was considered statistically significant.

**Table 1 cells-11-03378-t001:** Demographic Data.

**Patient Characteristics**	
*n*	32
Females	9
Age (years), mean ± SEM	68.5 ± 1.5
BMI (kg/m^2^), mean ± SEM	28.5 ± 0.8
Perfusion time (min), mean ± SEM	96.4 ± 1.8
Number of grafts, mean	4
**Preoperative Medications (%)**	
Acetylsalicylic acid	80
ACE-inhibitors/ARBs	47
β-blockers	88
Calcium channel blockers	19
Diuretics	38
Insulin	6
Nitrates	66
Oral anti-diabetics	22
Statins	72

SEM: standard error of mean; ACE: angiotensin converting enzyme; ARB: angiotensin receptor blocker.

**Table 2 cells-11-03378-t002:** Correlations between baseline (Base) and maximum (Max) YKL-40 levels and baseline levels, maximum levels and areas under curve (AUC) of monocyte chemotactic protein-1 (MCP-1), interleukin-8 (IL-8), 8-isoprostane and troponin T (TnT).

			MCP-1 (pg/mL)			IL-8 (pg/mL)			8-Isoprostane (pg/mL)			TnT (ng/L)	
			Base	Max	AUC	Base	Max	AUC	Base	Max	AUC	Base	Max
**YKL-40 ng/mL**	Base	r=	0.246	**0.398**	0.041	**0.500**	0.188	0.049	0.336	0.271	0.367	0.406	**0.374**
		*p*=	0.208	**0.040**	0.841	**0.007**	0.349	0.804	0.080	0.163	0.841	0.059	**0.050**
	Max	r=	**0.398**	0.119	**0.440**	0.306	0.278	0.245	0.349	0.018	0.225	0.096	**0.457**
		*p*=	**0.040**	0.561	**0.028**	0.121	0.169	0.218	0.075	0.928	0.269	0.724	**0.016**

Variables underwent logarithmic transformation to aim to obtain normal distributions. Pearson’s correlation coefficient was used for parametric and Spearman’s rank correlation coefficient for nonparametric variables. *p* < 0.05 was considered statistically significant (shown in bold).

## Data Availability

All data are included in the manuscript.

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
