# Peer review of "Inflammatory Glycoprotein YKL-40 Is Elevated after Coronary Artery Bypass Surgery and Correlates with Leukocyte Chemotaxis and Myocardial Injury, a Pilot Study"

_cells, 2022, doi:10.3390/cells11213378_

Round 1
Reviewer 1 Report
Thank you for providing an opportunity to review this manuscript. Although this research tried to establish an association of YKL-40 with myocardial injury, the sample size was too small to have such a conclusion. Several important covariates, such as age and sex differences, might influence this association. Therefore, since the outcome (myocardial injury is common), it is suggested that the authors can continue this research, add more samples and re-submit this manuscript. Authors are also suggested to do subgroup analysis, stratification, and interaction analysis.
Authors only presented/measured the YKL-40 up to 24 hrs. But they did not explain why they had taken samples only up to 24 hrs. Thus, they have been suggested that at least the injury is recovered (maybe 5-7days).
The authors concluded that “Thus YKL-40 may be an effector and biomarker of myocardial inflammation and injury and subsequent fibrosis following heart surgery”. In their discussion section, I recommend that the authors compare their findings with animal or cell line studies before concluding this study.
Reviewer 2 Report
The authors have analyzed YKL-40 and markers of myocardial injury (troponin T [TnT]), leukocyte activation (monocyte chemotactic protein-1 [MCP-1] and interleukin 8 [IL-8]) and oxidative stress (8-isoprostane) in blood samples undergoing elective CABG surgery. They found that YKL-40 may participate in the inflammatory cascade following open heart surgery utilizing CPB and function as a biomarker of inflammation and myocardial injury.
Comments:
1. A Table summarized the clinical characteristics of the patients has to be provided.
2. The lipid profile, body mass index, blood pressure, glycemia, drug therapy have to be reported.
3.In the introduction section at line 62, they reported that 32 patients were underwent elective CABG surgery but in the Methods four patients were excluded. Please correct it.
4. In the result section, they reported that during surgery, YKL-40 levels remained relatively constant. However, Figure 1 shows that the plasma levels of YKL-40, MCP-1, 91 IL-8 and 8-isoprostane significantly increased at the onset of cardiopulmonary bypass (CPB) in comparison of baseline ( BL). Please explain this point.
5. YKL-40, MCP-1, 91 IL-8 increased at 4 hours after onset of 121 CPB (CPB+4 h) but not 8-isoprostane levels. What are the physiological causes of these effects?
6. How is the level of troponin during surgery steps in comparison to baseline?
7. There is no a clear scientific evidence that KL-40 may participate in the inflammatory cascade following open heart surgery utilizing CPB.
Author Response
Please see attacment.

Round 2
Reviewer 2 Report
The paper is impoved and the authors have addressed to reviewer's comments